# Synthesis and Intramolecular Energy- and Electron-Transfer of 3D-Conformeric Tris(fluorenyl-[60]fullerenylfluorene) Derivatives

**DOI:** 10.3390/molecules24183337

**Published:** 2019-09-13

**Authors:** He Yin, Min Wang, Loon-Seng Tan, Long Y. Chiang

**Affiliations:** 1Department of Chemistry, University of Massachusetts Lowell, Lowell, MA 01854, USA; He_Yin@student.uml.edu (H.Y.); wangmin81@gmail.com (M.W.); 2Functional Materials Division, AFRL/RXA, Air Force Research Laboratory, Wright-Patterson Air Force Base, Dayton, OH 45433, USA; loon.tan@us.af.mil

**Keywords:** Tri[60]fullerenyl stereoisomers, *cis*-*cup*-form of 3D-stereoisomers, tris(diphenylaminofluorene), 3D-configurated nanostructures, intramolecular energy transfer for singlet oxygen production, intramolecular electron transfer for superoxide radical production

## Abstract

New 3D conformers were synthesized to show a nanomolecular configuration with geometrically branched 2-diphenylaminofluorene (DPAF-C_2M_) chromophores using a symmetrical 1,3,5-triaminobenzene ring as the center core for the connection of three fused DPAF-C_2M_ moieties. The design led to a class of *cis*-*cup*-tris[(DPAF-C_2M_)-C_60_(>DPAF-C_9_)] 3D conformers with three bisadduct-analogous <C_60_> cages per nanomolecule facing at the same side of the geometrical molecular *cis*-*cup*-shape structure. A sequential synthetic route was described to afford this 3D configurated conformer in a high yield with various spectroscopic characterizations. In principle, a nanostructure with a non-coplanar 3D configuration in design should minimize the direct contact or *π*-stacking of fluorene rings with each other during molecular packing to the formation of fullerosome array. It may also prevent the self-quenching effect of its photoexcited states in solids. Photophysical properties of this *cis*-*cup*-conformer were also investigated.

## 1. Introduction

Photoinduced intramolecular energy and electron transfer phenomena in organo [60]fullerene derivatives having a covalent molecular composition of both an electron donor and a [60]fullerenyl or nanocarbon acceptor components were demonstrated over a number of years [1,2,3]. The energy process may involve the facile triplet state of fullerene and other chromophores [4,5,6,7,8]. This type of nanomolecular system was used in many technological applications [9,10], including photovoltaic devices [11,12], sensors and switches [13], and photodynamic therapy [14,15]. Fullerene-based nanostructures with multiple C_60_ cages [16] in the structure were found to be suitable for the applications of nanocars [17,18,19], photoswitches [20], molecular heterojunctions [21], and catalysts [22]. Unusual molecular properties of multi-cage fullerene objects were theoretically predicted [23,24,25,26]. Recently, similar photophysical chemistry was also simulated in the modulation of photoswitchable dielectric properties to observe a large amplification of dielectric constants in a material combination form of multi-layered core-shell nanoparticles (NPs) [27,28,29]. The latter system was based on photoinduced intramolecular electronic charge-polarization of light-harvesting chromophoric nano[60]fullerenyl conjugates, such as 9,9-di(3,5,5-trimethylhexyl)-2-diphenylaminofluorenyl-methano[60]fullerene C_60_(>DPAF-C_9_) (**1**-C_9_, Figure 1). The polarization provided detectable dielectric property enhancement in a layered [60]fullerosome membrane structure on gold-shelled nanoparticles. The phenomena were based on the high electronegativity of C_60_> cage making it possible to rapidly shift an electron from light-harvesting DPAF (diphenylaminofluorene) donor moiety within the molecular structure to the C_60_> moiety. This resulted in the formation of the corresponding charge-separated (CS) state C_60_^−^·(>DPAF-C_9_)^+^ as the source of polarized charges. In fact, ultrafast concurrent intramolecular energy and electron transfer kinetics within **1**-C_9_ were substantiated previously by femtosecond transient absorption measurements (pump-probe) [30,31,32]. When these negative charges were distributed, delocalized, and stabilized in the fullerosome membrane array at the shell layer on core-shell NPs, it resulted in a CS state with a lifetime prolonged enough for the detection of dielectric characteristics. The process involved interlayer photoinduced plasmonic energy transfer from the Au shell layer to the outer shell layer of C_60_(>DPAF-C_9_) in addition to the fact that **1**-C_9_ itself is also photoresponsive and excitable under light irradiation.

One crucial parameter to consider is the method of molecular packing within the fullerosome shell layer. In this regard, strong tendency of light-harvesting chromophores to aggregate among *π*-conjugated planar aromatic moieties can cause either concentration-dependent self-quenching effects of excited states or luminescence in the solid phase, including fullerosome. The *π*–*π* stacking may result in significant reduction of many photophysical properties. This type of packing aggregation can be partially minimized by the use of highly bulky and geometrically hindered *π*-conjugated chromophores in a structural design to restrict or distort intramolecular rotation bonding units with steric hindrance [33]. In the case of DPAF-C_9_, we recently developed and synthesized highly restricted 3D conformers based on inter-connected three DPAF-C_9_ chromophore units giving a structure of tris(DPAF-C_9_) (**2**-C_9_) [34] to prevent and minimize the tendency of planar DPAF units to undergo aromatic–aromatic stacking, overlapping, and aggregation via intermolecular hydrophobic–hydrophobic interactions in solid thin-films. This structural modification led the enhancement of photophysical properties, including the intensity of photoluminescence (PL) and electroluminescence (EL) emissions [35].

Accordingly, we extended the similar structural strategy to design new 3D conformers tris[(DPAF-C_2M_)-C_60_(>DPAF-C_9_)] (**4**-C_2M–9_), as shown in Figure 1, for the study. The stereochemical modification was based on the construction of 3D geometrically branched tris[(DPAF-C_2M_) (**2**-C_2M_) chromophore having a shared central benzene unit among three 2-diphenylaminofluorenes. Highly steric hindrance at the corresponding 1,3,5-phenylfluorenylaminobenzene moiety forced three fluorene ring moieties to twist either upward or downward away from the central benzene plane with a large torsional angle on the nitrogen atom. This resulted in the formation of three to four possible stereoisomeric configurations, such as *cis*-*cup*-**2**-C_2M_, *trans*-*chair*-**2**-C_2M_, and *cis/or trans*-*propeller*-**2**-C_2M_, as shown in Figure 1. All of these conformer forms were proposed to be capable of fully eliminating the tendency of **4**-C_2M–9_ in inducing *π*–*π* aromatic–aromatic type stacking packing and to allow all C_60_ > cages to interact with each other via strong hydrophobic–hydrophobic interaction forces between (C_60_>)−(C_60_>) fullerenyl cages, forming the nano-layer array of fullerosome membrane.

## 2. Results and Discussion

Rapidly responsive nanophotonic physical properties of 3D conformers tris[(DPAF-C_2M_)-C_60_(>DPAF-C_9_)] (**4**-C_2M−9_) are achievable by specifically associating a donor–acceptor type chemical structure to a photomechanism having the ability to create a largely enhanced intramolecular energy and electron transfer efficiency. This mechanism occurs between C_60_> acceptor and DPAF-C_2M_/and DPAF-C_9_ donor moieties bonded on fullerenes. In our functional group design of **4**-C_2M−9_, a methanoketo bridging unit was used to trigger the keto-enol isomerization tautomerism that is capable of inducing *π*-periconjugation between C_60_> and DFAP to provide a partial conjugation pathway for enhancing the *π*-electron mobility around the conjugated system of molecular nanostructures [36,37]. In addition, the new molecular design of stereoisomeric tris(fluorenylphenylamino)benzene [tris(DPAF-C_9_)] analogous was proven to act as a fluorophore showing high intensity of photoluminescence and electroluminescence emission efficiency [34,35]. This revealed the successful utilization of stereochemistry to allow hindered and branched 3′,5′,5′-trimethylhexyl (C_9_) arms to maintain the space-separation of three planar DPAF moieties intramolecularly within the nanostructure. It also behaved similarly in intermolecular packing that improved light-harvesting efficiency.

Based on the molecular formation energy based density functional theory (DFT) calculations of three plausible stereoisomers of tris(DPAF-C_9_) (**2**-C_9_) via B3LYP/6-31G* level of theory using SPARTAN08 [34,35], the results revealed high stability of the *cis*-*cup*-form with other forms in stability order of *cup* > *chair* > *propeller*, as shown in Figure 1. This agreed well for the alkyl *n*-C_6_, *n*-C_7_, and *n*-C_8_ substituents owing to the influence by strong dispersion interactions within the alkyl chains. In the case of the methyl and the ethyl substituents, *trans*-*chair*-form may have been more stable than *cis*-*cup*-form. Accordingly, three C_4_-analogous substituents of tris(DPAF-C_2M_) (**2**-C_2M_) facing toward each other in the 3D molecular space above the central benzene ring should have brought in the minimum alkyl–alkyl interaction forces required to keep a slight favor of the *cis*-*cup*-form over either *trans*-*chair*- or *cis*/*trans*-*propeller* form.

Synthetically, the precursor molecule 2-bromo-9,9-bis(methoxyethyl)fluorene (BrF-C_2M_) was prepared by alkylation of 2-bromofluorene with mesylated methoxyethanol reagent using potassium *t*-butoxide as a base in a high yield of 90% (Figure 1). The key 3D-conformeric intermediate **2**-C_2M_ was synthesized by the reaction of BrF-C_2M_ with 1,3,5-tris(phenylamino)benzene (TPAB) in the presence of sodium *t*-butoxide, a catalytic amount of tris(dibenzylideneacetone)dipalladium(0) [Pd_2_(dba)_3_(0)], and *rac*-2,2′-bis(diphenylphosphino)- 1,1′-binaphthyl (*rac*-BINAP, 0.75 mol%) in anhydrous toluene at reflexing temperature for a period of 72 h to yield 82% of the product as a light yellow solid after chromatographic purification [SiO_2_, hexane–ethylacetate (1:1, *v*/*v*) as the eluent]. Subsequent attachment of three C_60_(>DPAF-C_9_) (**1**-C_9_) on **2**-C_2M_ should have led to the nanostructure of tris[(DPAF-C_2M_)-C_60_(>DPAF-C_9_)] (**4**-C_2M–9_) having all **1**-C_9_ moieties extended outward from the central 1,3,5-triaminobenzene core. Prior to the attachment of three **1**-C_9_ on **2**-C_2M_, it was functionalized by the Friedel–Crafts acylation at C7 position of 2-diphenylaminofluorene moiety [36] with excessive *α*-bromoacetyl bromide (11.4 equiv.) in the presence of aluminum chloride (17 equiv.) in 1,2-dichloroethane at 0–10 °C to ambient temperature overnight to afford the corresponding *α*-bromoacetylfluorene derivative in this intermediate step of reactions. It resulted in viscous yellow semi-solids in 48% yield of tris(BrDPAF-C_2M_) (**3**-C_2M_). It was purified by either column or thin-layer chromatography (TLC) [silica gel, hexane–EtOAc (1:1, *v*/*v*) as eluent, *R*_f_ = 0.5 on TLC]. The final step of synthesis for the preparation of 3D-conformers **4**-C_2M–9_ was performed by the reaction of **3**-C_2M_ with **1**-C_9_ (5.0 equiv.) in the presence of 1,8-diazabicyclo[5.4.0]undec-7-ene (DBU) in toluene at room temperature for 8.0 h. During the first Bingel reaction period of 2.0 h, mono- and bis-adducts could be observed and detected by the TLC technique, showing two brown fraction bands (*R*_f_ = 0.2 and 0.3 on TLC plate, corresponding to mono- and bis-adducts, respectively). Subsequently, three brown fraction bands (*R*_f_ = 0.2, 0.3, and 0.4) appeared after 4 h of reaction, indicating sequential additions of C_60_(>DPAF-C_9_) on the starting substrate tris(BrDPAF-C_2M_) (**3**-C_2M_) with the *R*_f_ band at 0.2 gradually disappeared. At the end of the reaction, only two bands (*R*_f_ = 0.3 and 0.4) remained, with the latter being identified as a major product band of **4**-C_2M–9_. After purification of this fraction by column chromatography (silica gel) using toluene–ethyl acetate (7:3) as the eluent, the brown solids of tris[(DPAF-C_2M_)-C_60_(>DPAF-C_9_)] were obtained in 79% yield.

The compound **4**-C_2M–9_ exhibited good solubility in common organic solvents owing to its possession of three DPAF-C_9_ (with a total of six branched C_9_-alkyl groups) and three DPAF-C_2M_ (with a total of six methoxyethyl groups) moieties (Figure 1), making three C_60_> cages become encapsulated in the center of the 3D molecular configuration. The 3D configuration of *cis*-*cup* form resulted in these alkyl groups being the main structural moieties interacting with the solvent. Accordingly, the compound had solubility (20 mg/mL in CHCl_3_ or CH_2_Cl_2_) over 10 times higher than that of C_60_ itself in toluene (1.4 mg/mL).

### 2.1. Spectroscopic Characterization of Synthetic 3D Configurated Fullerenyl Nanomaterials

All chemical conversions of intermediate chemicals to the corresponding products at each step of the reactions were characterized by various spectroscopic techniques. The functional attachment of three *α*-bromoaceto groups (3.0 equiv.) to tris(DPAF-C_2M_) given the product of tris(BrDPAF-C_2M_) was verified by both infrared (FT-IR) and ^1^H-NMR spectra. The former showed a new strong carbonyl (–C=O) stretching absorption band centered at 1673 cm^−1^, indicating each of the three carbonyl groups being bonded on a phenyl moiety, such as that of **2**-C_2M_. This absorption wavelength was in clear contrast to the strong band absorption at 1725 cm^−1^ normally detectable for an alkyl carbonyl group. In the case of ^1^H-NMR spectrum, tris(BrDPAF-C_2M_) displayed characteristic new peak signals of three methylene protons (H_α_) next to the carbonyl group of the *α*-bromoaceto moiety at *δ* 4.49 (Figure 2Ab) as compared with that of **2**-C_2M_ (Figure 2Aa). Subsequent attachment of three C_60_(>DPAF-C_9_) moieties to each of the three DPAF moieties of **3**-C_2M_ with a cylopropylaceto bridging unit on each C_60_> cage of three **1**-C_9_ (applied as a reagent) showed evidence of changing solubility characteristics of the product **4**-C_2M–9_ matching with those of **1**-C_9_. Its FTIR spectrum displayed a slight shift of cyclopropyl keto group absorption band to *υ*_max_ 1679 cm^−1^ (Figure 3d), which was assigned to the carbonyl (C=O) stretching band. It was also accompanied by an olefinic (C=C) absorption band centered at *υ*_max_ 1591 cm^−1^. Both C=O and C=C bands were correlated to those of C_60_(>DPAF-C_9_) (Figure 3b) and tris[C_60_(>DPAF-C_9_)] (Figure 3c), showing a nearly identical absorption wavenumber. Most importantly, we were able to detect two typical fullerenyl cage bands at *υ*_max_ 574 (w) and 524 (s) cm^−1^ (Figure 3d). These two bands were corresponding characteristic absorptions used to provide evidence of (C_60_>)-related monoadducts and bisadducts with absorption wavenumbers and relative intensity ratios differentiable from those of C_60_ (Figure 3a) and **1**-C_9_ substituents (Figure 3b). Accordingly, we applied this IR technique for the product structure verification during the chemical conversion from **3**-C_2M_ to **4**-C_2M−9_. Upon conversion of C_60_ to its monoadducts, such as those of Figure 3b,c, the remaining cage structure of C_60_> exhibited the same two bands with a reduced peak intensity for the 574 cm^−1^ band. The intensity of this band was further reduced in the structure of <C_60_>-like bisadduct, such as **4**-C_2M−9_ (Figure 3d). Furthermore, the latter band at 524 cm^−1^ still remaining strong was indicative of successful attachment of C_60_(>DPAF-C_9_) moieties on **3**-C_2M_ due to the possibility of having the second malonate bridging unit being attached at or near the equator region of the C_60_> cage. This would have led to the retention of a C_60_ half-cage that enabled absorption at 524 cm^−1^.

By the analysis of ^1^H-NMR spectrum, disappearance of peaks corresponding to the chemical shift of *α*-proton (H_α_) on the α-bromoaceto group of **3**-C_2M_ at *δ* 4.49 (Figure 2Ab) along with the appearance of new peaks over *δ* 5.25–5.78 (Figure 2Ac) provided evidence of successful formation of a cyclopropanyl keto-bridging unit between a C_60_> cage and the fluorene moiety. By using the previously reported chemical shift values of the *α*-proton (H_α_’) in C_60_(>DPAF-C_9_) (Figure 2Ad) [20] and the related *α*-proton (H_α_″) in tris(DPAF-C_9_) (**2**-C_9_, Figure 2Ae) [23] as the reference, we assigned these proton peaks to a combination of H_α_ and H_α_’. A large downfield shift of the H_α_ chemical shift from those of **3**-C_2M_ at *δ* 4.49 to *δ* 5.25–5.78 for **4**-C_2M–9_ provided clear evidence of three C_60_(>DPAF-C_9_) moieties being attached on the corresponding α-bromoaceto bridging units of **3**-C_2M_. The characteristics of the multipeaks for H_α_ and H_α_’ revealed a less symmetrical environment among these six protons of **4**-C_2M–9_ as the geometric shape of the nanostructure extended to a 3D configuration. It is worthwhile to mention that a large down-fielded chemical shift value of either H_α_’ or H_α_’’ away from the normal value of *δ* 2.1–2.5 for an alkyl aceto-*α*-proton was caused by the influence of strong [60]fullerenyl ring current in close vicinity. In addition, the alkyl proton regions over *δ* 2.93 (methoxy proton, 18H) and *δ* 2.75–2.64 (ethylenoxy proton, 12H) of Figure 2Ac (marked by beige) were correlated well to those of **2**-C_2M_ at *δ* 2.91 (18H) and *δ* 2.64 (12H) (Figure 2Aa), respectively, indicating good retention of central tris(DPAF-C_2M_) core region without any structural change of methoxy groups during the Friedel–Crafts acylation reaction. It also showed a new group of methyl proton peaks at *δ* 0.30–2.07 having an integration ratio value of 113.28, which represented 114 fluorenyl protons of C_9_-alkyl proton (19Hs for each of the two C_9_-alkyls of DPAF-C_9_) and was indicative of six C_9_-alkyl groups in the structure, consistent with the product structure. Additional ^1^H-NMR spectroscopic data analyses on proton integrations of all proton peaks to substantiate and count for the molar quantity ratio among fluorene, methoxyethyl, and C_9_ alkyl moieties to prove the molecular formulation of **4**-C_2M–9_ are provided in Appendix A.

Most importantly, characteristics of central benzene protons at the core region could be used for the analysis of the relatively geometric configuration of three fluorenyl rings with respect to each other. With a symmetrical structure of TPAB, three benzene protons should have displayed a singlet peak in its ^1^H-NMR spectrum. Upon attachment of a bulky fluorenyl moiety at each diphenylamino group, it induced high torsional stress and steric hindrance at the nitrogen atom that forced each 9,9′-di(methoxyethyl)fluorene moiety to twist or rotate either upward or downward from the central benzene plane. The action resulted in two main 3D conformers: *cis*-*cup*-**2**-C_2M_ and *trans*-*chair*-**2**-C_2M_. The former with three C_60_(>DPAF-C_9_) moieties facing upward on the same side in the structure gave a singlet H_a_ peak (Figure 1). The latter with one facing downward and two C_60_(>DPAF-C_9_) moieties facing upward in the structure resulted in two proton peaks for *trans*-H_a_’ (1H) and *trans*-H_b_’ (2H) (Figure 2Ba). By analyzing Figure 2Aa of tris(DPAF-C_2M_), a sharp singlet proton peak at δ 6.53 was assigned to the chemical shift of central benzene proton *cis*-H_a_. This peak was compared with that of the H_a_ proton peak of *cis*-*cup*-tris(DPAF-C_9_) (*cis*-*cup*-**2**-C_9_, Figure 2Bb) showing even better resolution of the peak profile, indicating a high purity of one 3D conformer fraction in a *cis*-*cup*-**2**-C_2M_ form. Surprisingly, this conformer fraction was, in fact, the major product. Apparently, the hydrophobic–hydrophobic dispersion interaction forces derived from three methoxyethyl chains and heteroatoms were stronger than those among all C_4_-alkyl groups, which led to higher tendency in formation of the *cis*-*cup*-form. Accordingly, subsequent attachment of three C_60_(>DPAF-C_9_) moieties on *cis*-*cup*-**2**-C_2M_ led to a similar formation of corresponding *cis*-*cup*-tris[(DPAF-C_2M_)-C_60_(>DPAF-C_9_)] (*cis*-*cup*-**4**-C_2M–9_), all having **1**-C_9_ moieties facing upward from the central benzene core at the same side with respect to each other. Additional structural analyses and discussions are provided in Appendix A.

In the case of the potential formation of regio-isomers of **4**-C_2M−9_ at the <C_60_> moiety, since the monoadduct structure of C_60_(>DPAF-C_9_) was well-defined, with the assistance of X-ray single crystal structural analysis of C_60_(>DPAF-C_2_) [31,36], its attachment on tris(BrDPAF-C_2M_) (**3**-C_2M_) was believed to be governed by the bulkiness and the steric hindrance of both relatively large entities to result in only a limited number of region isomers on the C_60_ cage. This was proven by the ^1^H-NMR spectrum of **4**-C_2M−9_ showing only several H_α_ and H_α_’ proton peaks at roughly *δ* 5.2–5.8 (Figure 2Ac) instead of the broad band normally seen for the existence of a large number of region isomers. To our surprise, a peak at 524 cm^−1^ assigned the characteristic infrared absorption band of a half-C_60_ cage (as stated above) showed close resemblance to those of the monoadduct C_60_(>DPAF-C_9_) (Figure 3b) and tris[C_60_(>DPAF-C_9_)] (Figure 3c) at an identical wavenumber. This implied the structure of the major regio-isomeric products had both addend moieties located at the same half-sphere of a C_60_ cage that left the other half-sphere of C_60_ untouched.

### 2.2. Photophysical and Physical Properties of 3D Conformeric Fullerenyl Nanomaterials

Photophysical properties of the 3D conformer *cis*-*cup*-**4**-C_2M–9_ were compared with those of precursor intermediates using the UV-vis spectroscopic technique. They were governed by two photoresponsive moieties, including three electron (e^−^)-accepting fullerene cages and six light-harvesting DPAF antenna units as electron (e^−^)-donors. The use of the latter was to enhance the optical absorption capability at longer visible wavelengths. The absorption wavelength could be varied and modulated by the appropriate chemical modification of functional substituents on fluorenyl moiety to affect electron-pushing (donating) and pulling (accepting toward the molecular edge of C_60_> cage moiety) mobility across the molecular *π*-conjugation system. As shown in Figure 4Ae of *cis*-*cup*-**4**-C_2M–9_, optical absorption of C_60_> cage moieties appeared mainly at the broad band centered at 296 nm (1.82 × 10^5^ L mol^−1^ cm^−1^), whereas the band centered at 411 nm (1.11 × 10^5^ L mol^−1^ cm^−1^) was attributed to the absorption of DPAF moieties. Characteristics of the latter band were compared with those of tris(DPAF-C_2M_) (Figure 4Aa), tris(BrDPAF-C_2M_) (Figure 4Ab), *cis*-*cup*-tris[C_60_(>DPAF-C_9_)] (Figure 4Ac), and C_60_(>DPAF-C_9_) (**1**-C_9_, Figure 4Ad), showing a clear bathochromic shift of the 351 nm band of **2**-C_2M_ to 406 nm (7.91 × 10^4^ L mol^−1^ cm^−1^) of **3**-C_2M_, which matched roughly with the 404 nm band of **1**-C_9_ and the 402 nm band *cis*-*cup*-tris[C_60_(>DPAF-C_9_)] for the peak assignment. This assignment was also consistent with the observation of roughly equal absorption extinction coefficient (*ε*) values for **2**-C_2M_, **3**-C_2M_, and tris[C_60_(>DPAF-C_9_)] with the same three DPAF moieties per molecule. Upon the attachment of three C_60_(>DPAF-C_9_) moieties, optical absorptions of [60]fullerene moieties of *cis*-*cup*-**4**-C_2M–9_ (Figure 4Ae) at 296 nm became dominant in the spectrum with a higher *ε* value. It was accompanied by a weak characteristic (forbidden) steady state absorption band of the C_60_> moiety appearing at 692 nm (the insert of Figure 4A) with a slightly higher extinction coefficient for the monoadduct **1**-C_9_ than the bisadduct *cis*-*cup*-**4**-C_2M–9_, which was also consistent with the photophysical property discussion above and provided further confirmation of a conjugated fullerenyl nanostructure.

In addition, a roughly 2.1-fold higher *ε* value (1.11 × 10^5^ L mol^−1^ cm^−1^) of the 411 nm peak in Figure 4Ae compared to that of 404 nm band of *cis*-*cup*-tris[C_60_(>DPAF-C_9_)] was consistent with a double number of DPAF arms per molecule for the former. Furthermore, very efficient intramolecular energy transfer from the excited singlet state of DPAF-C_9_ antenna to C_60_> was detected, which nearly eliminated the fluorescence of C_60_(>DPAF-C_9_) (*λ*_ex_: 406 nm, Figure 4Bb). In high contrast, without any C_60_> cage in the structure, the compound of tris(DPAF-C_2M_) showed a strong intensity of fluorescence emission (*λ*_ex_: 352 nm, Figure 4Ba) that clearly indicated the loss of photoexcited energy being associated with the influence of [60]fullerene. With an additional DPAF-C_9_ antenna in the structure of **4**-C_2M–9_, it began to experience a slightly excessive fluorescence emission (*λ*_ex_: 410 nm, Figure 4Bc) after the majority of photoexcited DPAF-C_2M_ energy underwent direct intramolecular energy transfer to the closely bonded [60]fullerene cage.

In investigating the plausibility of photoinduced intramolecular electron (e^−^)-transfer capability within the nanostructure of the 3D conformer *cis*-*cup*-tris[(DPAF-C_2M_)-C_60_(>DPAF-C_9_)] (*cis*-*cup*-**4**-C_2M–9_), we first investigated the unit character of redox potentials among all structural components, including the bisadduct-based <C_60_> cage and the DPAF-C_9_ moieties for comparison using the cyclic voltammetric (CV) technique. Several CV measurements were performed on the sample of *cis*-*cup*-**4**-C_2M–9_ in a solution of CH_2_Cl_2_ containing (*n*-butyl)_4_N^+^-PF_6_^−^ as the electrolyte and Pt as both the working and the counter electrodes and with Ag/AgCl as the reference electrode.

To deliver appropriate redox potential analyses and data interpretation, related CV characteristics of a C_60_-bisadduct of C_60_(>*t*-Bu-malonate)_2_ with a <C_60_> cage attached by two *t*-butylmalonate groups and the precursor compound C_60_(>DPAF-C_9_) were collected. They were performed under the same CV condition over cyclic oxidation and reduction voltages versus Ag/Ag^+^ from −2.0 to 2.0 V as those for *cis*-*cup*-**4**-C_2M–9_, as shown in Figure 5. As a result, it displayed one reversible oxidation (^1^*E*_ox_ of 1.51 V) reduction (^1^*E*_red_ of 1.32 V) cyclic wave with the first half wave oxidation potential (^1^*E*_1/2,ox_) of 1.42 V for the DPAF moieties of *cis*-*cup*-**4**-C_2M–9_ at positive voltages (Figure 5Ab,Bb). In the negative voltage region, its CV diagram displayed three reversible reductions at −0.34 (^1^*E*_red_), −0.82 (^2^*E*_red_), and −1.29 V (^3^*E*_red_) with the corresponding cyclic oxidation waves at −0.15 (^1^*E*_ox_), −0.44 (^2^*E*_ox_), and −0.96 (^3^*E*_ox_), respectively. These data corresponded to the first to the third half wave reduction potentials of −0.25 (^1^*E*_1/2_,_red_), −0.63 (^2^*E*_1/2_,_red_), and −1.12 V (^3^*E*_1/2_,_red_), respectively. By comparison of these values to those of C_60_(>*t*-Bu-malonate)_2_ (Figure 5Aa,Ba) for the fullerene cage moiety and those of C_60_(>DPAF-C_9_) (Figure 5Ac,Bc) for both DPAF and fullerene cage moieties, highly consistent and reproducible redox potential characteristics among structural components were found that also substantiated the structural derivatization of tris[(BrDPAF-C_2M_) (**3**-C_2M_) with triple C_60_(>DPAF-C_9_) to form *cis*-*cup*-**4**-C_2M–9_. Accordingly, the latter exhibited combined CV characteristics of <C_60_> and DPAF-C_9_. These CV characteristics were reproducible for four repeated redox cycles with the reductive C_60_> and the oxidative DPAF potential profiles showing only slight changes at the potential range of −2.0 to 2.0 V. This implied good stability of the material under CV conditions that led to possible reuse of **4**-C_2M–9_.

### 2.3. Evidence of Intramolecular Energy- and Electron-Transfer Events within cis-cup-4-C_2M–9_ by Detection of Corresponding Reactive Oxygen Species (ROS)

There is an appropriate approach to substantiate intramolecular energy and electron transfer events within the nanomolecular structure of *cis*-*cup*-**4**-C_2M–9_ by directly detecting the photoinduced production of reactive oxygen species (ROS). In general, the most common ROS includes singlet oxygen (^1^O_2_) produced by the Type-II photomechanism via the intermolecular transfer of triplet energy to molecular oxygen (O_2_) and superoxide radical (O_2_^−^**·**) generated by the intermolecular transfer of electron (e^−^) to O_2_. For the former case, upon photoexcitation at the C_60_> cage moiety of *cis*-*cup*-**4**-C_2M–9_, the singlet excited state of bis-methanofullerenyl ^1^(<C_60_>)^*^ may undergo facile intersystem crossing in a nearly quantitative efficiency to its triplet excited state ^3^(<C_60_>)^*^ that can be accounted for by the efficient production of ^1^O_2_ via Type-II triplet energy transfer processes. Alternatively, if the photoexcitation process is aimed at either the DPAF-C_2M_ or the DPAF-C_9_ moiety of *cis*-*cup*-**4**-C_2M–9_, the resulting corresponding singlet excited states of either ^1^(DPAF-C_2M_)^*^ or ^1^(DPAF-C_9_)^*^ may undergo both pathways of (1) intramolecular energy transfer from either ^1^(DPAF-C_2M_)^*^ or ^1^(DPAF-C_9_)^*^ to the <C_60_> moiety to produce neutral DPAF-C_2M_ or DPAF-C_9_ and ^1^(<C_60_>)^*^, respectively; (2) intramolecular electron (e^−^)-transfer from either ^1^(DPAF-C_2M_)^*^ or ^1^(DPAF-C_9_)^*^ to the <C_60_> moiety to produce cationic (DPAF-C_2M_)^+^**·** or (DPAF-C_9_)^+^**·** and (<C_60_>)^−^**·**, respectively. Both events of (1) and (2) can occur concurrently. Subsequent intermolecular e^−^-transfer from (<C_60_>)^−^ to O_2_ produces the corresponding neutral <C_60_> and O_2_^−^ following the Type-I photomechanism.

Accordingly, by the direct detection of ROS on either ^1^O_2_ and/or O_2_^−^**·** upon irradiation on *cis*-*cup*-**4**-C_2M–9_ at either <C_60_> or DPAF-C_2M_/DPAF-C_9_ moiety, we were able to provide the evidence of intramolecular energy and electron transfer processes happening within this 3D-conformer. We selected two reliable fluorescent (FL) probes for the detection of either ^1^O_2_ or O_2_^−^·separately in the solution of *cis*-*cup*-**4**-C_2M–9_ with high selectivity and specificity as a crucial measure. To detect the former ROS ^1^O_2_, a synthetic highly fluorescent compound *α*,α’-(anthracene-9,10-diyl)-bis(methylmalonic acid) (ABMA) was used as the probe in the experiment. Its UV-vis absorption and fluorescence emission spectra are given in Figure 6a,b, respectively. In the probe reaction, chemical trapping of ^1^O_2_ by highly fluorescent ABMA resulted in the formation of non-fluorescent 9,10-endoperoxide product ABMA-O_2_ (Figure 7A). This chemical conversion allowed us to follow the intensity loss of fluoresce emission upon photoexcitation. The loss could be associated with the proportional quantity of ^1^O_2_ produced. The correlation was valid owing to a higher reaction kinetic rate of the trapping process in solution than the internal decay of ^1^O_2_ in the same solvent system of a DMF–CHCl_3_ (1:9, *v*/*v*) mixture. Experimentally, the quantity of ^1^O_2_ generated was monitored and counted by the relative intensity decrease of fluorescence emission (*λ*_em_) of ABMA at 428 nm under excitation wavelength (*λ*_ex_) of 380 nm. At this excitation wavelength, it matched partially with the optical absorption band of DPAF moieties of *cis*-*cup*-**4**-C_2M–9_ that led to a slight fluorescence emission (Figure 6c) after the intramolecular energy and the e^−^-transfer processes. It gave a slightly higher count in the overall FL intensity during the experiment (Figure 6d). In a typical probe reaction, a master solution of ABMA in DMF was diluted by CHCl_3_ prior to the addition of *cis*-*cup*-**4**-C_2M–9_. It was followed by periodical illumination using a light emitting diode (LED) lamp of white light (a power output of >2.0 W) operated at two major emission peak maxima (*λ*_max_) centered at 451 and 530 nm. The former light emission spectrum exhibited a sufficient bandwidth covering the 410–470-nm region for photoexcitation of DPAF moieties with optical absorption bands covering 380–500-nm (Figure 4Ae). As a result, we were able to detect rapid production of ^1^O_2_ by *cis*-*cup*-**4**-C_2M–9_ upon irradiation in a decreasing curve profile over a period of more than 120 min (Figure 7Ab).

The FL probe experiments were calibrated by a blank control run using the same probe concentration of ABMA alone and an illumination time scale in the absence of *cis*-*cup*-**4**-C_2M–9_ (Figure 7Aa). Apparently, we observed slight photodegradation of ABMA itself. This may have implied the existence of a photoinduced triplet state of ABMA in a low quantity due to exposure to short wavelength regions of the light emission bandwidth covering over ~380 nm of ABMA absorption bands.

In the case of detecting superoxide radical as the second ROS, a synthetic O_2_^−^**·**-reactive fluorescent probe precursor molecule, non-fluorescent potassium bis(2,4-dinitrobenzenesulfonyl)-2′,4′,5′,7′-tetrafluorofluorescein-10′ (or 11′)-carboxylate (DNBs-TFFC), was applied for the experiment. Its molecular structure was synthetically modified from that reported previously [38], showing good reaction selectivity with a high O_2_^−^**·**/^1^O_2_ sensitivity ratio. Since DNBs-TFFC itself is photodegradable, a dialysis film with the molecular weight cut-off (MWCO) of 100−500 Daltons was applied to hold the solution of *cis*-*cup*-**4**-C_2M–9_ in toluene–DMSO (9:1, *v*/*v*). The sack bag was separated from the solution of the probe DNBs-TFFC. The latter was kept in a cuvette with stirring during the fluorescence emission measurement. Only the solution of *cis*-*cup*-**4**-C_2M–9_ in the dialysis membrane sack was subjected to the white LED light exposure. Any superoxide radical produced was allowed to rapidly diffuse into the probe solution through the dialysis membrane and initiate the desulfonylation of DNBs-TFFC. The O_2_^−^**·**-trapping reaction led to the elimination of two dinitrobenzenesulfonyl moieties and yielded the corresponding bisphenol intermediate, as shown in Figure 7B. Rearrangement of the bisphenol intermediate to the ring-opening of lactone afforded highly fluorescent potassium 2′,4′,5′,7′-tetrafluorofluorescein-10′ (or 11′)-carboxylate regiosisomers (TFFC). The latter compound gave the fluorescence emission at 530 nm (*λ*_em_) with the excitation at 484 nm (*λ*_ex_). As the probe DNBs-TFFC was not a fluorescent compound, detected emission photon counts were fully associated with the quantity of TFFC produced. Measured total emission intensity counts were then correlated to the relative quantity of O_2_^−^**·** generated. As shown in Figure 7Bb, nearly linear progressive increase of fluorescence intensity counts over the full irradiation period was observed that revealed the constant production of O_2_^−^**·** from the photoexcited *cis*-*cup*-**4**-C_2M–9_. As discussed above, continuous irradiation on six DPAF moieties of *cis*-*cup*-**4**-C_2M–9_ by white LED light (2.0 W) stimulated photoexcitation from the ground to the singlet excited state. Subsequent intramolecular e^−^-transfer from ^1^(DPAF-C_n_)^*^ to <C_60_> moieties resulted in the formation of anionic [60]fullerenyl bisadduct radical (<C_60_>)^−^ intermediate. In the presence of O_2_, it was followed by further e^−^-transfer from (<C_60_>)^−^ intermediate to O_2_ to produce O_2_^−^ in a sequential multiple-step Type-I photomechanism. These results clearly provided the evidence of photoinduced intramolecular e^−^-transfer mechanism within the 3D-conformer *cis*-*cup*-**4**-C_2M–9_.

Furthermore, one of the main crucial criterion for the use of these types of C_60_-(light-harvesting antenna)_n_ conjugates, such as **4**-C_2M–9_, as the nano-photosensitizers for antibacterial inactivation (aPDI) is their high photostability. Unlike the conventional organic chromophore-based photosensitizers suffering rapid photodegradation, C_60_-(light-harvesting antenna)_n_ based nano-drugs were found to be capable of a single dose with multiple aPDI/PDT (photodynamic therapy) treatments [39,40,41,42].

## 3. Experimental Section

### 3.1. Chemicals and Reagents

Reagents of sodium *t*-butoxide, *α*-bromoacetyl bromide, aluminum chloride (AlCl_3_), 1,8-diazabicyclo[5.4.0]undec-7-ene (DBU), *rac*-2,2′-bis(diphenylphosphino)-1,1′-binaphthyl (BINAP), and tris(dibenzylideneacetone)dipalladium(0) [Pd_2_(dba)_3_(0)] were purchased from Aldrich Chemicals, Natick, Massachusetts, USA and used without further purification. Fullerene materials with a purity of 99% were purchased from Suzhou Dade Carbon Nanotechnology Co., Ltd. Suzhou, Jiangsu, China. Anhydrous grade solvent of tetrahydrofuran (THF) was used and further dried via refluxing over sodium and benzophenone overnight and distilled under reduced pressure (10^−1^ mmHg). The precursor compounds including 1,3,5-tris(*N*-phenylamino)benzene (TPAB) and 2-bromo-9,9-di(methoxyethyl)fluorene (BrF-C_2M_) were synthesized according to our previous procedures [34].

### 3.2. Instruments for Spectroscopic Measurements

^1^H-NMR spectra were recorded on either Bruker and Spectrospin Avance 500 or Bruker AC-300 spectrometer. UV-vis spectra were recorded on a PerkinElmer *Lambda 750* UV spectrometer. Fluorometric traces were collected using a PTI QuantaMaster^TM^40 Fluorescence Spectrofluorometer. The light source used in this experiment included a collimated white LED light with an output power of 2.0 W (Prizmatix, Southfield, MI, USA). Infrared spectra were recorded as KBr pellets on a Thermo Nicolet *AVATAR 370* FTIR spectrometer. Cyclic voltammetry (CV) was record on EG&G Princeton Applied Research 263A Potentiostat/Galvanostat using Pt metal as the working electrode, Ag/AgCl as the reference electrode, and Pt wire as the counter electrode at a scan rate of 10 mV/s. The solution for CV measurements was prepared in a concentration of 1.0–5.0 × 10^−3^ M in appropriate solvents containing the electrolyte Bu_4_N^+^-PF_6_^−^ (0.1 M).

### 3.3. Synthesis of N^1^,N^3^,N^5^-Tris(9,9-di(methoxyethyl)fluoren-2-yl)-1″,3″,5″-tris(phenylamino)-benzene as Tris(DPAF-C_2M_) (**2**-C_2M_)

Synthetic procedure for the preparation of tris(DPAF-C_2M_) was slightly modified from those methods reported recently [34]. In general, a mixture of BrF-C_2M_ (7.33 g, 20.3 mmol, excess), TPAB (1.16 g, 3.3 mmol), and sodium *t*-butoxide (1.94 g, 20.3 mmol) was dissolved in anhydrous toluene (75 mL) and stirred for 1 h to give a homogeneous solution. The catalyst Pd_2_(dba)_3_(0) (0.023 g, 0.25 mol%) and *rac*-BINAP (0.046 g, 0.75 mol%) were added to the solution, followed by heating to refluxing temperature under nitrogen for a period of 72 h. After cooling the resulting mixture to room temperature, it was washed with water three times by extraction, the organic layer was separated, and it was dried over sodium sulfate. After solvent evaporation, a small quantity of crude paste was tested on the TLC plate to show the major product at *R*_f_ = 0.6 using hexane–ethylacetate (1:1, *v*/*v*) as the eluent. This product spot had a dense yellow-brown color on the top accompanied by a light visible tail. The tail portion was assumed to be the product in the *trans-chair* form. This tail portion was subsequently separated from the main top portion of the *cis-cup* form via column chromatography, followed by the TLC plate purification using silica gel as the stationary phase and hexane-ethylacetate (1:1, *v*/*v*) as the eluent to afford the product of *cis*-*cup*-tris(DPAF-C_2M_) (**2**-C_2M_) as light yellow solids in 82% yield (3.23 g). Verification of the *cis*-*cup* form was based on the detection of a singlet proton peak at *δ*6.53 (H_a_) corresponding to three central core 1,3,5-benzene protons, whereas the trans-chair form gave two proton groups at a integration ratio of 2:1. Spectroscopic data: FT-IR (KBr) *υ*_max_ 3062 (w, aromatic C-H stretching), 3031 (w), 3016 (w), 2969 (w, aliphatic C-H stretching), 2922 (m), 2875 (m), 2814 (w), 1571 (s, C=C), 1491 (s, anti-symmetric deformations of CH_3_ groups and scissor vibrations of CH_2_ groups), 1448 (s), 1428 (m), 1381 (w, symmetric deformations of CH_3_ groups), 1292 (m, asymmetric stretching vibrations of C-N-C), 1248 (m, asymmetric stretching vibrations of C-N-C), 1212 (w), 1176 (w), 1155 (w), 1110 (s, stretching vibrations of C-O-C), 1039 (w), 948 (w), 877 (w), 829 (w), 754 (m), 738 (s, out-of-plan deformation of C-H), 711 (m, out-of-plan deformation of C-H), 692 (s), and 511 (w) cm^−1^; UV-vis (CHCl_3_, 1.0 × 10^−5^ M) *λ*_max_ (*ε*) 321 (7.24 × 10^4^ L mol^−1^ cm^−1^) and 351 nm (7.66 × 10^4^ L mol^−1^ cm^−1^); ^1^H NMR (500 MHz, CDCl_3_) *δ* 7.58 (s, 3H, br), 7.52 (d, 3H), 7.35–7.24 (m, 9H), 7.15–7.03 (m, 18H), 6.89 (t, 3H), 6.53 (s, 3H, central benzene protons H_a_), 2.91 (s, 18H), 2.65 (m,12H), and 2.19 (m, 12H).

### 3.4. Synthesis of N^1^,N^3^,N^5^-Tris(7-α-bromoacetyl-9,9-di(methoxyethyl)fluoren-2-yl)-1″,3″,5″-tris-(phenylamino)benzene as Tris(BrDPAF-C_2M_) (**3**-C_2M_)

The compound *cis*-*cup*-tris(DPAF-C_2M_) (*cis*-*cup*-**2**-C_2M_, 0.53 g, 0.44 mmol) was added to a homogeneous suspension of AlCl_3_ (1.0 g, 7.5 mmol) in 1,2-dichloroethane (40 mL) at 0 °C with vigorous stirring. The reagent *α*-bromoacetyl bromide (1.0 g, 5.0 mmol) was added slowly over 10 min while maintaining the temperature between 0–10 °C. The mixture was then stirred overnight at room temperature. An excessive amount of AlCl_3_ remaining in the solution was quenched by slow addition of water (50 mL) while maintaining the temperature below 45 °C. After washing sequentially with dil. HCl (1.0 N, 50 mL) and water (50 mL × 2), the organic layer was separated and dried over sodium sulfate and then concentrated in vacuo to give the crude product as viscous yellow semi-solids. It was purified by column chromatography (silica gel) followed by thin-layer chromatography (TLC) using hexane–EtOAc (1:1, *v*/*v*) as eluent to afford *cis*-*cup*-tris(Br-DPAF-C_2M_) (*cis*-*cup*-**3**-C_2M_) (at *R*_f_ = 0.5 on TLC plate) in 48% yield (0.33 g). Spectroscopic data: FT-IR (KBr) *υ*_max_ 3062 (w, aromatic C-H stretching), 3031 (w), 3016 (w), 2969 (w, aliphatic C-H stretching), 2922 (m), 2875 (m), 1673 (s, C=O), 1571 (s, C=C), 1491 (s, anti-symmetric deformations of CH_3_ groups and scissor vibrations of CH_2_ groups), 1467 (s), 1448 (s), 1428 (m), 1388 (w), 1348 (w), 1282 (s), 1251 (s), 1195 (m), 1176 (w), 1110 (s, stretching vibrations of C-O-C), 1035 (w), 948 (w), 879 (w), 823 (m), 755 (m), 740 (s, C-H out-of-plan deformation), 715 (m, C-H out-of-plan deformation), 694 (s), 617 (w) and 538 (w) cm^−1^; UV-vis (CHCl_3_, 1.0 × 10^−5^ M) *λ*_max_ (*ε*) 310 (7.65 × 10^4^ L mol^−1^ cm^−1^) and 406 nm (7.91 × 10^4^ L mol^−1^ cm^−1^); ^1^H NMR (500 MHz, CDCl_3_) *δ* 7.98 (m, 3H), 7.58 (m, 3H), 7.34–7.20 (m, 9H), 7.11–6.98 (m, 18H), 6.96 (m, 3H), 6.56 (m, 3H, aromatic protons of central phenyl ring), 4.49 (m, 6H, *α*-proton next on C_61_), 2.91 (s, 18H. primary C_2M_ alkyl protons next to O-atom), 2.81–2.50 (centered at *δ* 2.66, m, 12H, secondary C_2M_ alkyl protons next to O-atom), 2.48–2.07 (centered at *δ* 2.16, m, 12H, C_2M_ alkyl protons next to the fluorene ring).

### 3.5. Synthesis of N^1^,N^3^,N^5^-Tris(7-(1,2-dihydro-1,2-methanofullerene[60]-61-carbonyl)-9,9-di(methoxyethyl)fluoren-2-yl)-1″,3″,5″-tris(phenylamino)benzene) as Tris[(DPAF-C_2M_)-C_60_(>DPAF-C_9_)] (**4**-C_2M–9_)

The reagent 1,8-diazabicyclo[5.4.0]undec-7-ene (DBU, 0.21 g, 1.38 mmol) was added slowly to a homogeneous mixture of C_60_>(DPAF-C_9_) (1.1 g, 0.97 mmol) and *cis*-*cup*-tris(BrDPAF-C_2M_) (*cis*-*cup*-**3**-C_2M_, 0.31 g, 0.20 mmol) in anhydrous toluene (700 mL). During the Bingel reaction for the first 2.0 h, the products of mono- and bis-adducts became visible by the TLC technique, showing two brown bands at *R*_f_ = ~0.2 and ~0.3, respectively, using a mixture of toluene–EtOAc (7:3, *v*/*v*) as eluent. After a longer reaction period of 4 h, three brown bands close to each other were observed at *R*_f_ = ~0.2, ~0.3, and ~0.4 with the former band progressively becoming faint. At the end of the reaction (8.0 h), only two latter bands remained at *R*_f_ = ~0.3 and ~0.4 with the latter as the major chromatographic fraction. At this stage, the reaction mixture was concentrated to a 10% volume and then precipitated from methanol (100 mL) to afford the crude product mixture, which was isolated by centrifugation. Further purification by column chromatography (silica gel) using toluene to a solvent mixture of toluene–EtOAc (7:3, *v*/*v*) as the eluent with sequential increments of increasing solvent polarity afforded *cis*-*cup*-tris[(DPAF-C_2M_)-C_60_(>DPAF-C_9_)]. It was further purified by TLC with isolation of only the narrow dense color fraction band to give brown solids of *cis*-*cup*-**4**-C_2M–9_ in 79% yield (0.74 g) (at *R*_f_ = ~0.4 on TLC). Spectroscopic data: FT-IR (KBr) *υ*_max_ 3062 (w, aromatic C-H stretching), 3031 (w), 3016 (w), 2952 (w, aliphatic C-H stretching), 2925 (m), 2852 (w), 1679 (s, C=O), 1591 (s, C=C), 1490 (s, anti-symmetric deformations of CH_3_ groups and scissor vibrations of CH_2_ groups), 1465 (s), 1419 (m), 1346 (w), 1315 (m), 1274 (s), 1238 (m), 1211 (s), 1170 (m), 1110 (s, stretching vibrations of C-O-C), 1072 (w), 1029 (w), 950 (w), 879 (w), 815 (m), 746 (s), 694 (s), 574 (w) and 524 (s, <C_60_>) cm^−1^; UV-vis (CHCl_3_, 1.0 × 10^−5^ M) *λ*_max_ (*ε*) 296 nm (1.82 × 10^5^ L mol^−1^ cm^−1^) and 411 nm (1.11 × 10^5^ L mol^−1^ cm^−1^); ^1^H NMR (500 MHz, CDCl_3_) *δ* 8.65–8.12 (m, 12H, fluorenyl protons next to the keto group), 7.91–7.48 (m, 12H, fluorenyl protons), 7.31−7.06 [m, 57H, 45 aminophenyl protons and 12 fluorenyl protons (2H for each fluorene ring) next to N-atom], 6.57 (m, 3H, aromatic protons of the central phenyl ring), 5.78–5.25 (m, 6H, *α*-proton next on C_61_), 3.01–2.81 (centered at *δ* 2.93) (m, 18H, primary C_2M_ alkyl protons next to O-atom), 2.81–2.50 (centered at *δ* 2.65, m, 12H, secondary C_2M_ alkyl protons next to O-atom), 2.48–2.07 (centered at *δ* 2.17, m, 12H, C_2M_ alkyl protons next to the fluorine ring), 2.07−1.18 (m, 12H, C_9_ alkyl protons next to the fluorene ring), 1.18–0.94 (centered at *δ* 1.14) (m, 6H, tertiary C_9_ alkyl protons), 0.94–0.30 (centered at *δ* 0.70) (m, 96H, primary and secondary C_9_ alkyl protons).

### 3.6. ROS Measurements Using singlet oxygen (^1^O_2_)-Sensitive Fluorescent Probe

The compound *α*,α’-(anthracene-9,10-diyl)bis(methylmalonic acid) (ABMA) was used as a fluorescent probe for singlet oxygen (^1^O_2_) trapping. The quantity of ^1^O_2_ generated was monitored and counted by the relative intensity decrease of fluorescence emission of ABMA at 428 nm under excitation wavelengths of 380 nm (*λ*_ex_). A typical probe solution was prepared by diluting a master solution of ABMA (1.0 × 10^−5^ M in DMF, 0.4 mL) with an amount 9-fold in volume of CHCl_3_ (3.2 mL) in a cuvette (10 × 10 × 45 mm). The solution was added by a pre-defined volume of tris[(DPAF-C_2M_)-C_60_(>DPAF-C_9_)] in CHCl_3_ (1.0 × 10^−5^ M, 0.4 mL), followed by periodic illumination using an ultrahigh power white light LED lamp (Prizmatix, operated at the emission peak maxima centered at 451 and 530 nm with the collimated optical power output of >2.0 W in a diameter of 5.2 cm). Progressive fluorescent spectra were taken on the PTI QuantaMaster^TM^ 40 Fluorescence Spectrofluorometer.

### 3.7. ROS Measurements Using Superoxide Radical (O_2_^−^·)-Sensitive Fluorescent Probe

A superoxide radical (O_2_^−^∙)-reactive fluorescent probe, non-fluorescent potassium bis(2,4-dinitrobenzenesulfonyl)-2′,4′,5′,7′-tetrafluorofluorescein-10′ (or 11′)-carboxylate regioisomers (DNBs-TFFC), was used for the experiment. A typical probe solution [10^−6^ M in toluene–DMSO (9:1)] was prepared by diluting a stock solution of DNBs-TFFC in DMSO (10^−5^ M, 1.0 mL) by 10 times with toluene (9.0 mL). A dialysis film (CE) with the molecular weight cut-off (MWCO) of 100–500 Da was used to separate the solution of tris[(DPAF-C_2M_)-C_60_(>DPAF-C_9_)] [10^−6^ M in toluene–DMSO (9:1)] from the probe solution kept in a cuvette with stirring during the fluorescent measurement. Only the solution of **4**-C_2M_ in the membrane sack was subjected to the LED light exposure at the excitation wavelength of 400–700 nm (white light). The quantity of O_2_^−^**·** generated was counted in association with its reaction with DNBs-TFFC that resulted in the product of highly fluorescent potassium 2′,4′,5′,7′-tetrafluorofluorescein-10′ (or 11′)-carboxylate regioisomers (TFFC) with fluorescence emission at 530 nm (*λ*_em_) upon excitation at 484 nm (*λ*_ex_). The detected intensity increase of fluorescence emission was then correlated to the relative quantity of O_2_^−^ produced.

## 4. Conclusions

Previous studies on detected photoswitchable dielectric amplification phenomena by the simulated ferroelectric-like capacitor design using the construction and the fabrication of a fullerosome shell layer on core-shell *γ*-FeO_x_@AuNPs were based on the photoinduced intramolecular charge-polarization of (C_60_> acceptor)-(DPAF-C_n_ donor) conjugates [16–18]. The corresponding formation of dielectric ion-radical components (C_60_>)^−^ and DPAF^+^**·**-C_n_ within a fullerosome array layer was the basis of observed dielectric properties. Accordingly, several such conjugates were developed by the extension from the initial C_60_(>DPAF-C_9_) to demonstrate the correlation of the structural relationship and the chemical modifications to the enhanced dielectric properties, as stated above. Two interesting modifications both involved 3D-conformeric C_60_(>DPAF-C_9_) derivatives by fusing three phenyl rings of three diphenylamino groups together to form a shared central benzene moiety as the base of 3D configuration design. Specifically, the successful synthesis of *cis*-*cup*-tris[(DPAF-C_2M_)-C_60_(>DPAF-C_9_)] stereoisomer may be beneficial for use as positive (DPAF-C_n_)^+^ and negative charge (<C_60_>)^−^ carriers in enhancing photoinduced dielectric characteristics [43]. It can also be applied as the precursor building block in the synthesis of several C_60_- and C_70_-based ultrafast photoresponsive nonlinear two-photon absorptive nanomaterials. Accordingly, we demonstrated efficient intramolecular energy and electron transfer capabilities of 3D conformer *cis*-*cup*-**4**-C_2M–9_ using photophysical measurements and its effective production of singlet oxygen (via the energy transfer mechanism) and superoxide radicals (via the electron transfer mechanism). They can be applied as nano-photosensitizers [39,40,41,42] and nonlinear photonic agents [30,31,32,44].

## Figures and Tables

**Figure 1 molecules-24-03337-f001:**
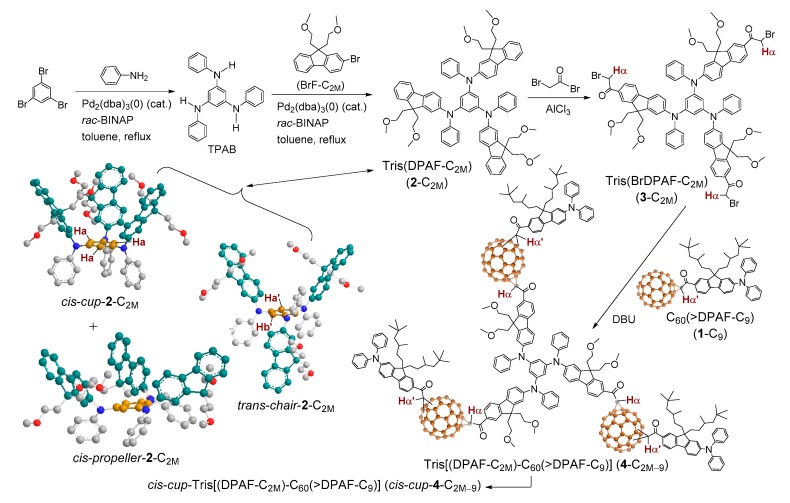
Synthetic methods for the preparation of 3D conformers of tris[(DPAF-C_2M_)-C_60_(>DPAF-C_9_)] (**4**-C_2M–9_) with reaction reagents given and three perspective 3D-configurations of the key stereoisomeric intermediate **2**-C_2M_.

**Figure 2 molecules-24-03337-f002:**
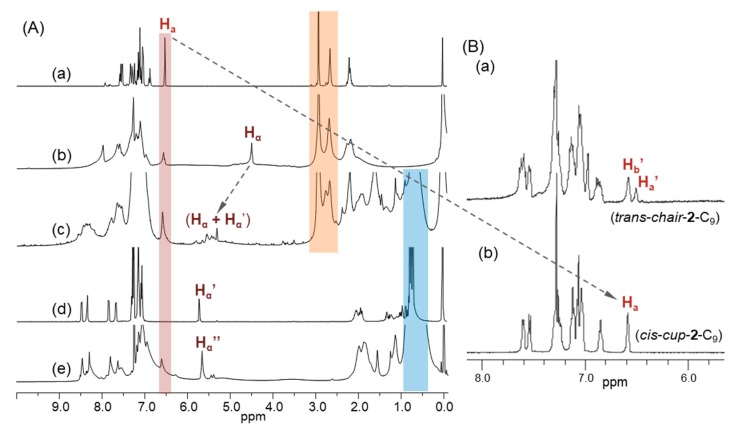
^1^H-NMR spectra (CDCl_3_) of (**A**) (a) tris(DPAF-C_2M_) (**2**-C_2M_), (b) tris(BrDPAF-C_2M_) (**3**-C_2M_), (c) tris[(DPAF-C_2M_)-C_60_(>DPAF-C_9_)] (**4**-C_2M−9_), (d) C_60_(>DPAF-C_9_) (**1**-C_9_), and (e) tris[C_60_(>DPAF-C_9_)]. (**B**) ^1^H-NMR spectra (CDCl_3_) of (a) *trans*-*chair*-tris(DPAF-C_9_) (*trans*-*chair*-**2**-C_9_) and (b) *cis*-*cup*-tris(DPAF-C_9_) (*cis*-*cup*-**2**-C_9_) for comparison.

**Figure 3 molecules-24-03337-f003:**
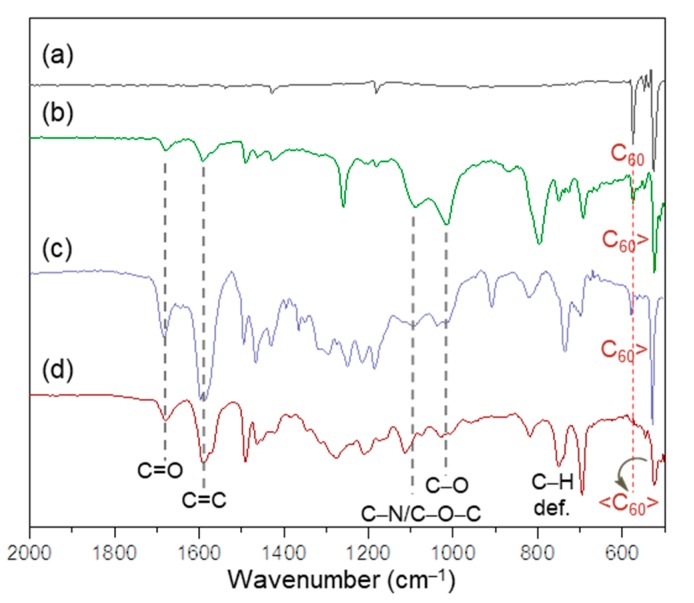
Infrared spectra (KBr) of (a) C_60_, (b) C_60_(>DPAF-C_9_) (**1**-C_9_), (c) tris[C_60_(>DPAF-C_9_)]. and (d) tris[(DPAF-C_2M_)-C_60_(>DPAF-C_9_)] (**4**-C_2M−9_).

**Figure 4 molecules-24-03337-f004:**
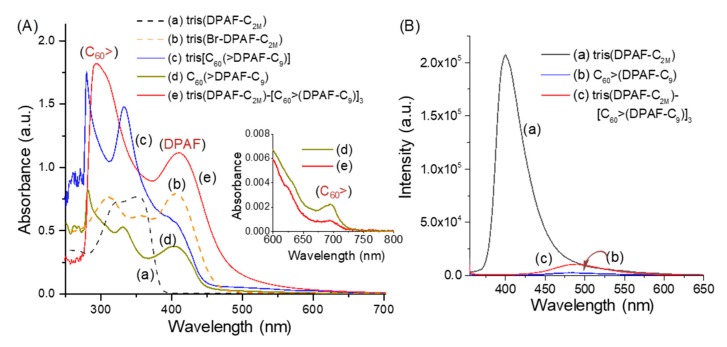
UV-vis spectra of (**A**) (a) tris(DPAF-C_2M_) (**2**-C_2M_), (b) tris(BrDPAF-C_2M_) (**3**-C_2M_), (c) *cis*-*cup*-tris[C_60_(>DPAF-C_9_)], (d) C_60_(>DPAF-C_9_) (**1**-C_9_), and (e) *cis*-*cup*-**4**-C_2M–9_, where (a), (b), and (e) were taken in CDCl_3_ and (c) and (d) were taken in toluene. (**B**) Fluorescence spectra of (a) tris(DPAF-C_2M_) (*λ*_ex_: 352 nm), (b) C_60_(>DPAF-C_9_) (*λ*_ex_: 406 nm), and (c) *cis*-*cup*-**4**-C_2M–9_ (*λ*_ex_: 410 nm). The concentration of all samples is 1.0 × 10^−5^ M.

**Figure 5 molecules-24-03337-f005:**
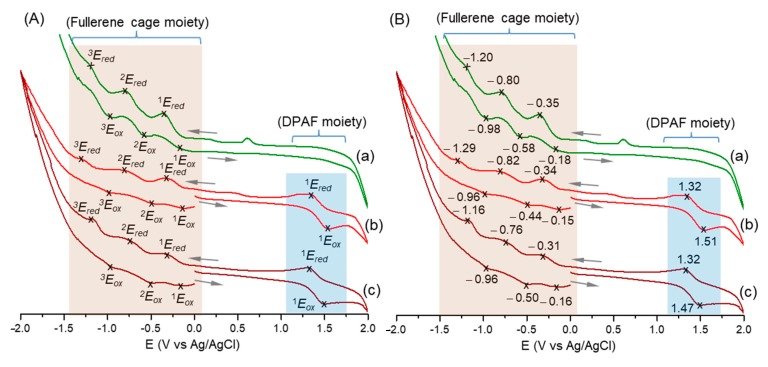
Cyclic voltammograms (CV) of (a) C_60_(>*t*-Bu-malonate)_2_, (b) *cis*-*cup*-tris[(DPAF-C_2M_)-C_60_(>DPAF-C_9_)], and (c) C_60_(>DPAF-C_9_), where (**A**) displays the sequential redox cycles of each compound with the assignments and (**B**) shows the corresponding potential voltage values at either the peak maximum or minimum of each redox cycle. All solutions were in a concentration of 5.0 × 10^−3^ M in CH_2_Cl_2_ using (*n*-butyl)_4_N^+^-PF_6^−^_ as the electrolyte (0.1 M), Pt as working and counter electrodes, and Ag/AgCl as the reference electrode at a scan rate of 10 mV/s.

**Figure 6 molecules-24-03337-f006:**
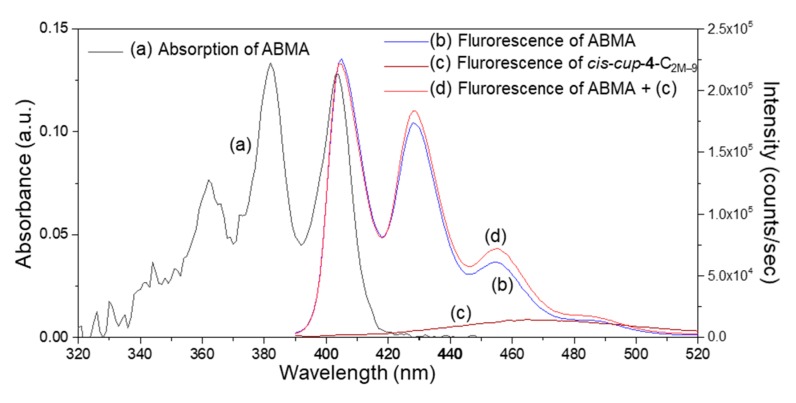
(a) UV-vis spectra of *α*,α’-(anthracene-9,10-diyl)bis(methylmalonic acid) (ABMA) and fluorescence (FL) emission spectrum of (b) ABMA, (c) *cis*-*cup*-tris[(DPAF-C_2M_)-C_60_(>DPAF-C_9_)] (*cis*-*cup*-**4**-C_2M–9_), and (c) a combination of ABMA and *cis*-*cup*-**4**-C_2M–9_ in a solvent mixture of DMF–CHCl_3_ (1:9, *v*/*v*) in a concentration of 10^−6^ M.

**Figure 7 molecules-24-03337-f007:**
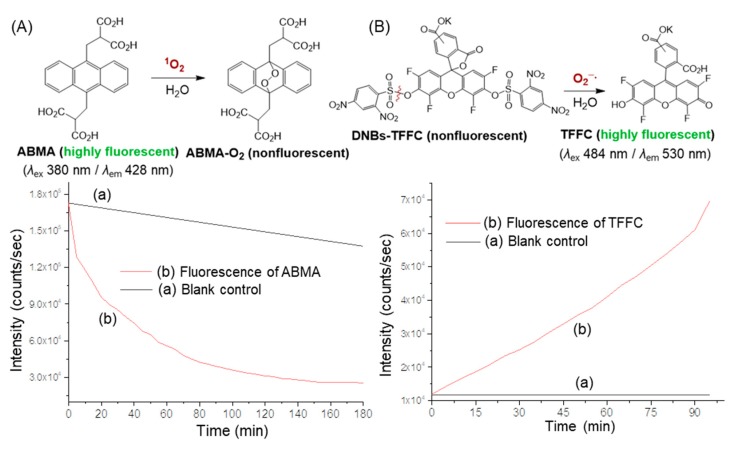
Fluorescence (FL) emission spectra of (**A**) ABMA and (**B**) tetrafluorofluorescein-10′ (or 11′)-carboxylate (TFFC) to correlate directly the singlet oxygen (^1^O_2_) and superoxide radical (O_2_^−^**·**) production efficiency, respectively, with (a) blank control and (b) a mixture of corresponding probe and *cis*-*cup*-**4**-C_2M–9_ samples in DMF–CHCl_3_ (1:9, *v*/*v*) at a concentration of 1.0 × 10^−5^ M using ABMA as the ^1^O_2_ trapping agent at *λ*_ex_ 350 nm and *λ*_em_ 428 nm and DNBs-TFFC as the O_2_^−^**·**-acceptor agent at *λ*_ex_ 484 nm and *λ*_em_ 530 nm for detection with the irradiation of white light emitting diode (LED) light.

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
