# Peer review of "Synthesis and Intramolecular Energy- and Electron-Transfer of 3D-Conformeric Tris(fluorenyl-[60]fullerenylfluorene) Derivatives"

_molecules, 2019, doi:10.3390/molecules24183337_

Round 1

Reviewer 1 Report

This manuscript reports the synthesis of a series of energy/electron donor/C60 polyads, and the study of the photophysical properties, the photosensitizing of the singlet oxygen and superoxide radical anions. The results are interesting, I recommend acceptance of the manuscript after minor revision.

Line 45, 'femtosecond pulse irradiation measurement' can be revised to 'femtosecond transient absorption measurements (pump-probe)';

Something is wrong in Figure 5; The Figures are not displayed correctly;

In order to confirm the energy transfer, the fluorescence lifetime changes of the energy donor should be studied;

Some related articles should be cited, Dyes and Pigments 2019, 171, 107756; J. Org. Chem., 2012,77, 5305−5312; Chem. Eur. J. 2009, 15, 7382 – 7393; Some review articles on triplet photosensitizers can be also cited: Chem. Soc. Rev., 2013, 42 (12), 5323–5351; Chem. Soc. Rev., 2013, 42, 77-88.

Author Response

The response to the reviewer 1's comments is provided in the attached pdf file.

Reviewer 2 Report

The manuscript entitled “Synthesis and Intramolecular Energy- and Electron-2 Transfer of 3D-Stereoisomeric Tris(fluorenyl-3 [60]fullerenylfluorene) Derivatives” describe the synthesis of large molecules containing several fullerene’s moieties along with fluorine bridges. The authors study and discuss physical properties of the obtained material, namely energy and electron transfer. The manuscript gives rise to a plethora of questions and if some of them are excusably minor, there are several crucial inconsistencies undermining the whole experimental part.

A. The presented synthetic scheme (Figure 1) does not leave a very good impression for the following reasons:

A1) It is rather unclear why the Friedel-Crafts acylation (3rd reaction) has such specific regioselectivity. This reaction was previously published only by the authors and the previously presented yield does not correspond to the yields presented in the current manuscript. (previously, the authors have shown 50% yields for the single acylation, now the yield does not alter, although there are three consequent acylation now). Nevertheless, the major problem here is a poor characterization of the product (3-C2M), the authors present only 1H NMR and IR, which are, by no means, sufficient for an unambiguous structure determination. At least 13C should be provided.

A2) The reaction 4, where 1-C9 interacts with 3-C2M. First of all, it is not explained what exact isomer of 1-C9 was exploited. Moreover, the chirality of 1-C9 is not mentioned, was it one of the enantiomers or racemate mixture? Thus, both stereo- and regio- aspects remain unclear. Consequently, neither regio- nor stereo- selectivity of the reaction are clear for the product (4-C2M-9). There are three fullerene’s moieties, each can be attached through [6,6] or [6,5], there are also six chiral centers. Having all this in mind, it becomes rather intriguing what exactly was obtained as a final and actually studied product. As it is currently presented, any reader would come to conclusion that it is a mixture of a hundred of isomers rather than individual substance. Therefore, the authors should clarify the final constitution of their product. Otherwise, the value of the physical studies is rather questionable. It is also not clear which isomer was used for calculations.

Moreover, I am very uncomfortable with the way how the authors exploit certain terms. For example:

i) 3D-steroisomer, stereoisomers are inherently three dimensional objects (except for probably alkenes). So the addition of this “3D” is rather questionable (the prefix "stereo-" already means "three-dimensional"), especially when there is no such terminology (at least to the best of my knowledge).

ii) The so-called “3D-stereoisomers”: “cis-cup”, “trans-chair” and “cis-propeller” (Figure 1) are actually conformers. If the rotation barriers are high enough (which is not a case here) they should be called rotamers.

Due to mentioned above major issues, I do not see any point in listing further minor flaws. Thus, I would not recommend this manuscript for publication in “Molecules”, unless at least the mentioned above major points are clarified and discussed.

Author Response

The response to the comments of the reviewer 2 is provided in the attached pdf file.

Author Response

The response to the comments of the reviewer 3 is provided in the attached pdf file.

Round 2

Reviewer 3 Report

The authors have resolved all my questions but I am not sure about the concerns of other reviewers. Nevertheless, I recommend the manuscript in the present version.

As for me, I consider the object of the study (tri-cage fullerene derivatives) is very complex. Thus, we should not expect flawless works in this field and the weak spots of the study may be resolved in the continuation of the present work.